# Nutritional Prognosis of Patients Submitted to Radiotherapy and Its Implications in Treatment

**DOI:** 10.3390/nu16091363

**Published:** 2024-04-30

**Authors:** Mariana Maroso Molina Irigaray, Lidiani Figueiredo Santana, Arnildo Pott, Valter Aragão do Nascimento, Rita de Cássia Avellaneda Guimarães, Albert Schiaveto de Souza, Karine de Cássia Freitas

**Affiliations:** 1Graduate Program in Health and Development in the Central-West Region of Brazil, Federal University of Mato Grosso do Sul, Campo Grande 79079-900, Brazil; valter.aragao@ufms.br (V.A.d.N.); rita.guimaraes@ufms.br (R.d.C.A.G.); albert.souza@ufms.br (A.S.d.S.); karine.freitas@ufms.br (K.d.C.F.); 2State University of Mato Grosso do Sul (UEMS), Dourados 79804-970, Brazil; lidi_lfs@hotmail.com; 3Laboratory of Botany, Institute of Biociências, Federal University of Mato Grosso do Sul, Campo Grande 79070-900, Brazil; arnildo.pott@gmail.com

**Keywords:** nutritional assessment, oncology, gastrointestinal neoplasms, head and neck neoplasms

## Abstract

Oncological patients show intense catabolic activity, as well as a susceptibility to higher nutritional risk and clinical complications. Thus, tools are used for monitoring prognosis. Our objective was to analyze the nutrition prognosis of patients who underwent radiotherapy, correlating it with outcomes and complications. We performed a retrospective transversal study based on secondary data from hospital records of patients who started radiotherapy between July 2022 and July 2023. We established Prognostic Scores through a combination of Prognostic Nutritional Index (PNI) and a Subjective Global Assessment (SGA), assessed at the beginning and end of treatment. Score 3 patients, with PNI ≤ 45.56 and an SGA outcome of malnutrition, initially presented a higher occurrence of odynophagia, later also being indicative of reduced diet volume, treatment interruption, and dysphagia. SGA alone showed sensitivity to altered diet volume, dysphagia, and xerostomia in the second assessment. Besides this, PNI ≤ 45.56 also indicated the use of alternative feeding routes, treatment interruption, and hospital discharge with more complications. We conclude that the scores could be used to indicate complications; however, further studies on combined biomarkers are necessary.

## 1. Introduction

Cancer is a disease characterized by the disordinate and uncontrollable multiplication of cells with mutated DNA [1]. It is considered a severe public health problem [2], with estimates for 2050 of 35.3 million cases of cancer worldwide and a mortality rate of 18.5 million people [3]. Individuals with cancer naturally show a weakened nutritional state due to the greater energy expenditure they display in response to the metabolic stress they suffer [1]. This factor is exacerbated when it comes to cancers of the head and neck (HNC) and gastrointestinal tract (GIT), and is strongly related to nutritional risk factors, since the process of chewing, swallowing, digesting, and absorbing nutrients is compromised [4,5,6]. In 2022, there was an incidence of more than 4.3 million cases of cancer of the GIT and HNC varieties worldwide, resulting in more than 2.4 million deaths [7].

In addition to the deleterious effects of cancer, treatment therapies, such as radiotherapy, can lead to clinical and metabolic changes such as nausea, vomiting, diarrhea, odynophagia, and dysphagia, contributing to a reduction in the amount of energy ingested, which further compromises the nutritional status of patients [8,9]. Malnourishment has a direct impact on the progression of the disease, with patients tending to present greater toxicities and a shorter tolerance time to treatment, which culminates in a lower therapeutic response, as well as a worsening quality of life and reduced survival rate [9,10]; thus, malnutrition in cancer patients is a common and serious concern, which significantly reduces the prognosis of patients [10].

As malnutrition is a common complication in cancer patients and often accelerates the progression of the disease, affecting treatment outcomes [11], studies have been carried out on pre-surgical patients with the associated application of nutritional assessment models to identify clinical and nutritional prognosis, such as the Patient-Generated Subjective Global Assessment (PG-SGA), and predictors of complications have been elaborated based on laboratory tests, such as the Prognosis Nutritional Index (PNI) [4,12,13,14,15]. On the other hand, there is a gap in the literature on studies involving this investigation, with the application of indices for assessing nutritional status and prognosis, as well as a lack of studies on patients undergoing non-surgical treatment, such as radiotherapy, which has a strong impact on nutritional status and can generate complications and/or worsen clinical conditions [16].

Considering this situation, it will be informative to analyze the prognoses of patients submitted for radiotherapy in a university hospital based on the markers PNI and PG-SGA, in association with (or isolated from) a bad nutritional status, in order to determine its relation to complications in the treatment. Thus, in addition to the studies already undertaken, it is possible to in this way identify nutritional risk and anticipate development complications, allowing early nutritional intervention and multi-professional management, effective before beginning treatment and throughout, assuring better clinical and nutritional recovery.

## 2. Materials and Methods

### 2.1. Study Design

We undertook a transversal retrospective study, surveying data from records of cancer patients who started treatment at the Radiotherapy Unit of the University Hospital Maria Aparecida Pedrossian (UFMS), Campo Grande (MS), Brazil, from July 2022 to July 2023.

### 2.2. Sampling

Sampling was undertaken by consulting the electronic medical records of patients 18 years of age or more, of both genders, who started treatment for cancer in the gastrointestinal tract (GIT) and head and neck (HN) regions, including accessory glands and other compartments that interfere with the digestive system (tonsils, larynx, trachea) in the radiotherapy outpatient unit of Humap, UFMS, Brazil (Figure 1).

We excluded records of patients who we could not contact or who did not consent to the use of their data, as well as teenagers who had started treatment in the sampling period and those affected by tumors in other areas than those specified.

The project was approved by the Ethics Committee in Research with Human Beings of the Fundação Universidade Federal do Mato Grosso do Sul (UFMS), under the CAAE number 60894522.8.0000.0021 and assent register 5.570.138.

### 2.3. Tools and Procedures of Data Collection

We analyzed the medical records of patients, obtaining information about age, gender, cancer site, number of radiotherapy sessions carried out, and the absorbed ionizing radiation dose, programmed in gray (Gy). A stratification of the number of treatment sessions carried out was established, considering the data collected, which ranged from 10 to 35 sessions (containing 3 classifications: less than or equal to 20 sessions, 21 to 29 sessions, and greater than or equal to 30 sessions). In relation to the radiation dose, we considered that the radiation received ranged from 30 to 70 Gy, with daily doses of 1.5 to 3 Gy/day 5 times a week, and the average dose received was 55.9 Gy. Along with the dose values used in other studies [17,18,19,20], a radiation dose cut-off point of 60 Gy was established.

Furthermore, information was sought on the patient’s weight, or anthropometry, as well as the percentage of weight loss during treatment and the nutritional diagnosis using the Patient-Generated Subjective Global Assessment (PG-SGA), developed specifically for cancer patients [21,22]. In addition, serum albumin and total lymphocyte values were collected for application in the Prognostic Nutritional Index (PNI), according to Onodera [23]. The ideal cut-off point for PNI was 45.56, obtained using the Characteristic Receptor Operation Curve (ROC), based on the results of the PNI and SGA.
PNI=(10 × serum albumin)+(0.005 × total count of lymphocytes)

We developed a punctuation score that compiled prognostic markers from both PNI and SGA, as presented in Table 1. We applied the prognostic tests at the start and end of treatment, according to the availability of laboratory test and SGA equipment, corresponding to the first and second assessments.

Finally, we identified the primary outcomes for patients, aside from the occurrence of complications throughout the treatment, as alterations in diet consistency, alterations in ingested volume, the change of feeding to an alternative route, hospital admission, and treatment interruption (with a suspension of more than 3 days being considered pertinent for the study), with consideration of the findings in the literature [24,25]. In addition, the development of mucositis, dermatitis, dysphagia, odynophagia, xerostomia, and diarrhea was assessed, with consideration of the presence or absence of symptoms in the analyses. Among the outcomes, we focused on death and hospital discharge without complications, with less than three complications, and with three or more complications.

### 2.4. Data Analyses

The cut-off point for PNI was determined via the ROC curve using the program MedCalc version 22.016, utilizing the method of Hanley and McNeil (1982) [26] and considering a 5% significance level. The evaluation of the association between measured variables and the Prognostic Score in the first and second assessments was checked using the chi-square test, with Bonferroni correction when necessary, given the need to test multiple comparisons between proportions. The same test was also utilized in evaluating the association between the same variables and the classification of the Subjective Global Assessment (SGA) or the Prognostic Nutritional Index (PNI). We presented the other results as descriptive statistics or in the form of tables and graphs. The statistical analysis was performed using the statistical program SPSS, version 24.0, considering a 5% significance level.

## 3. Results

First, we determined an ideal cut-off point for the Prognostic Nutritional Index (PNI) of 45.56, with a *p* = 0.018. The PNI varied from 30.16 to 60.6 (mean of 43.59), with an area under the curve (AUC) of 0.664 (IC = 0.567 to 0.752), a significance of 78.9% (IC = 54.4 to 93.9%), and a specificity of 61.1% (IC = 50.3 to 71.2) (Figure 2).

Based on the data obtained, we identified 67 patients who underwent radiotherapy treatment from July 2022 to July 2023, in whom we observed a predominance of the elderly, with a mean age of 61.59 years (59.7%), as well as males (74.6%) and those undergoing concomitantly chemotherapy (74.6%) (Table 2).

Among the cancer sites studied (head and neck and gastrointestinal tract), we identified a higher number of cases of the pharynx (28.35%), covering the nasopharynx, oropharynx, and hypopharynx. Another prevalent site was the rectum (23.9%), in addition to the anal canal region (7.46%), comprising 31.36% of cancer cases.

Besides this, we observed a higher percentage of patients with Score 1, i.e., who had a PNI above 45.56 but an SGA suggesting malnutrition. The same was observed for other individuals, as only 28.4% of the patients had PNI scores indicative of a bad prognosis, while 61.2% were diagnosed with malnutrition using SGA, be it moderate or severe.

Correlating the prognostic score that the patient presented with the degree of cancer, we verified a significant difference (31.6%, *p* = 0.018) in patients of Prognostic Score 2 with stage 3 cancer compared to those with the other scores. We also observed that of 19 patients diagnosed with stage 3 cancer, 79% had some nutritional impairment (Scores 1, 2 and 3) (Table 3). We saw the same in patients with stage 4 cancer, with 44.8% having a diagnosis of malnutrition by SGA (Prognostic Score 1), and in 24.1%, besides malnutrition, we also saw a PNI below 45.56 (Prognostic Score 3), totaling 68.9% of patients with impaired nutritional status and prognosis (Table 3).

Despite the lack of significant difference, we observed 50 patients undergoing radiotherapy concomitant with chemotherapy; 77% had some impaired nutrition, i.e., Prognostic Score 1, 2 or 3. Similarly, in patients with Score 0 (well-nourished), around 47.1% underwent only radiotherapy (Table 3). Besides this, we determined that patients with nutritional consequences presented higher percentages of weight loss occurrence than those with Score 0. Around 75.6% of patients (Score 1, 2 and 3) lost up to 5% of their weight, while the same occurred in only 24.3% of individuals with Score 0. This difference was more notable in patients with over 10% weight loss, and 80% showing this effect had a weakened nutritional status, while 20% were well-nourished (Table 3).

Concerning treatment outcome, despite also showing no significant difference, we observed a higher percentage without complications (33.3%) in Score 0 patients than in the others. Furthermore, among the 18 patients who finished radiotherapy with 3 or more complications, around 72.2% showed a weakened nutritional status (Table 3).

Having analyzed the complications presented in the first assessment (Table 4), we observed, contrary to what was expected, that Score 2 patients had a lower incidence of dysphagia than the others (*p* = 0.006). Furthermore, we identified a trend in the higher occurrence of mucositis (*p* = 0.065), odynophagia (*p* = 0.082), and diarrhea (*p* = 0.080) in patients with Prognostic Score 3 (Table 4).

Despite the lack of significant difference, we also observed that the use of an alternative feeding route (72.7%), hospital admission (83.3%), treatment interruption (82.7%), dermatitis (73.5%), dysphagia (79.4%), and odynophagia (100%) represented higher percentages in patients with weakened nutritional status (Score 1, 2 and 3) (Table 4). We also identified in well-nourished patients with Prognostic Score 0 a higher percentage with no hospital admission during radiotherapy (30.9%) and no treatment interruption (36.8%). Besides this, among symptoms indicating a nutritional impact, patients with Score 0 presented lower rates of dermatitis (26.5%), dysphagia (20.6%), and odynophagia (0%) (Table 4).

When addressing the correlations between complications, indicated by the markers of PNI and PG-SGA, in isolation (Table 5), we observed that patients with a diagnosis of malnutrition via PG-SGA (B and C) had higher rates of dysphagia (65.9%, *p* = 0.002) than the well-nourished patients (26.9%). We also noted a trend towards less odynophagia (*p* = 0.064) and diarrhea (*p* = 0.065) in well-nourished patients.

The other alterations and outcomes did not reach significance in terms of their correlation with nutritional status. In spite of this, we identified that patients with SGA-A presented higher percentages of no alteration in food consistency (73.1%), the lower use of alternative feeding routes (88.5%), fewer hospital admissions (88.5%), and a lower degree of treatment interruption (61.5%) (Table 5).

Among the symptoms of nutritional impact, well-nourished patients showed a higher degree of absence of mucositis (65.4%), odynophagia (100%), xerostomia (76.9%), and diarrhea (73.1%). In the moderately and severely undernourished, we observed higher percentages of alteration in diet volume (70.7%) and dermatitis development (78%) (Table 5).

In the PNI, contrary to expectations, we identified a higher occurrence of alterations in volume of food ingested in patients with PNI > 45.56 (*p* = 0.030), as well as a trend towards less xerostomia (*p* = 0.066) (Table 5). Similar to SGA, in the PNI, correlations in the other alterations and outcomes did not present significant values in the first assessment. Furthermore, we determined a notably higher percentage of treatment interruption (57.9%) and dermatitis (57.9%) in patients with PNI ≤ 45.56. In patients with PNI above the cut-off, we identified higher percentages of no hospital admission (83.3%) and no treatment interruption (62.5%), and among the symptoms, the near absence of odynophagia stands out (95.8%) (Table 5).

We identified 42 patients who had undergone blood tests at the start and end of treatment, allowing for the assessment of both PNI and the related scores. In this group, we observed an increase of 42.5% in the number of patients showing Score 3, from 19.4% in the first assessment to 61.9% in the second assessment (Figure 3).

The second highest alteration was seen in the number of patients with a prognostic score of 1, which was reduced by 25.1% (the first assessment showed around 41.8% of patients with this score, which reduced to 16.7%). We also observed a reduction of 16.5% in patients with Score 0, from 28.4% at the treatment’s start to 11.9% at its end. The number of patients with a score of 2 remained balanced during the assessments (Figure 3).

Among 42 individuals, we verified that the initial score worsened in 25, indicating an alteration in more than half of the reassessed patients (59.5%). Besides this, we observed around 26 undernourished patients according to SGA, and 6 patients had PNI ≤ 45.56 in the first assessment, while in the second assessment, this number increased to 33 undernourished according to SGA, and 26 patients’ PNI scores were reduced.

Initially, 14.3% of patients (n = 6) presented PNI ≤ 45.56 and malnutrition via SGA at the same time. In the second assessment, this value increased to 61.9% (n = 26), an increase of 47.6%. We verified that the SGA presented greater sensitivity at both assessments to malnutrition, developing from 61.9% to 78.6% (Figure 4). We also noted that, of the 26 patients showing a worsened PNI at the second time point, around 21, i.e., 91.3%, were already showing some degree of malnutrition as diagnosed by PG-SGA at the first assessment.

In the second assessment of the prognostic scores, this time considering the complications presented, we observed that patients with a score of 3 showed higher degrees of alteration in the volume of diet ingested (71%, *p* = 0.028), as well as treatment interruptions (83.3%, *p* = 0044) and the development of dysphagia (81.5%, *p* = 0.005). In the same way, we can state that patients with a score of 0, i.e., well-nourished, suffered less of an alteration in their ingested volume (3.2%, *p* = 0.028) and a lower development of dysphagia (3.7%, *p* = 0.005) (Table 6).

We also detected a trend towards treatment outcomes with three or more complications in patients with a prognostic score of 3 (92.3%, *p* = 0.059), as well as higher percentages of well-nourished patients, classified with score 0, finishing radiotherapy without complications (33.3%, *p* = 0.059).

These correlations did not present significant values, but we observed high percentages of patients with Score 3 requiring alternative feeding routes (90%) and hospital admission (83.3%), besides manifesting odynophagia (100%) and xerostomia (81.8%). Among patients with a weakened nutritional status, i.e., Scores 1, 2, and 3, we identified a high percentage of adaptation to diet consistency (92.3%) and the development of dermatitis (93.9%) (Table 6).

In the second assessment of isolated markers, derived from SGA and PNI, regarding the complications shown, we observed that undernourished patients, classified B and C using SGA, suffered significantly greater alterations in food volume (81.8%, *p* = 0.024), in addition to the development of dysphagia (75.8%, *p* = 0.003) and xerostomia (33.3%, *p* = 0.044), compared to the well-nourished patients. We also saw a trend towards higher levels of occurrence of dermatitis (84.8%, *p* = 0.058) in patients with an SGA score showing malnutrition (Table 7).

The other correlations between nutritional status determined by SGA and complications were non-significant. However, well-nourished patients (SGA-A) also showed fewer alterations in ingested food consistency (11.1%), besides fewer cases of alternative feeding routes (11.1%) and hospital admission (11.1%) being required, as well as treatment interruptions (33.3%), odynophagia (0%), and diarrhea (11.1%) (Table 7).

Regarding the Prognostic Index, individuals with PNI ≤ 45.56 showed greater alterations in ingested diet volume (83.3%, *p* = 0.026), in addition to a higher percentage of use of alternative feeding routes (33.3%, *p* = 0.022), treatment interruption (53.3%, *p* = 0.03) and the development of dysphagia (76.7%, *p* = 0.008) (Table 7). Similarly, patients with indices above 45.56 had a degree of protection against such alterations.

Furthermore, considering the outcomes regarding the numbers of complications at discharge, patients with PNI > 45.56 exhibited greater numbers of hospital discharge without complications during treatment (33.3%, *p* = 0.029). In turn, individuals with PNI ≤ 45.56 concluded their treatment with three or more complications (40%, *p* = 0.029) (Table 7). Despite non-significant results regarding the other correlations, we observed a lower percentage of hospital admission in patients with PNI > 45.56 (91.7%), as well as a lack of odynophagia (100%) and xerostomia (83.3%). In patients with PNI ≤ 45.56, we observed a higher percentage of dermatitis development (80%) (Table 7).

## 4. Discussion

Cancer is still classified as a severe public health problem, and is the second leading cause of death in the world [2,3]. It is known that cancer is a highly catabolic disease that compromises nutritional status [27,28]; it is thus extremely important to pay attention to nutritional status using nutritional and prognostic markers in order to anticipate and improve the efficiency of cancer care [4,10,29,30].

The assessment of our sample evidenced a prevalence of the elderly (59.7%) and males (74.6%), as well as cancers located in the head and neck region (67.14%). Our findings employed global statistics, showing that the incidence of head and neck cancer is 2.8 times higher in men than in women [7], and that this vulnerability is associated with exposure to risk factors such as excessive alcohol and tobacco consumption [2,31]. In relation to the ages of patients, other studies have also identified higher percentages of elderly people affected by the disease [5,32,33,34,35], and there is also an association between greater nutritional risk and the development of malnutrition, to the detriment of adults [36,37].

We verified that out of 67 patients, 61.2% had already presented malnutrition according to PG-SGA at the beginning of treatment, 46.3% having moderate malnutrition and 14.9% severe, while at the treatment’s end, 78.6% were undernourished (57.14% moderate and 21.42% severe). In the same way, concerning the prognostic scores, significant nutritional deterioration was observed, with a 42.5% increase in the prevalence of Score 3 between the assessments (19.4% in the first and 61.9% in the second), and a reduction in well-nourished patients (Score 0).

Studies around the world have shown that malnutrition in oncology is associated with the site of the disease, the nutritional symptoms the patient presents, the time and dose of treatment used, as well as concomitant therapies [18,19,20]. In relation to the oncological site, it has been observed that patients with cancer in the head and neck region present a greater degree of nutritional risk, which can affect up to 90% of patients [38,39], especially when the pharyngeal region is affected [39,40]. In the present study, malnutrition was observed in 75.5% of HN patients at the start of treatment, rising to 88.88% after reassessment, and of the 45 patients with cancer in this region, 28.35% of cases affected the pharyngeal region.

With regard to the other conditions that increase the likelihood of treatment toxicity and malnutrition, we also observed that 47.8% underwent 21 to 29 radiotherapy sessions, and 40.3% had 30 sessions or more. Besides, in 74.6%, the treatment was chemotherapy, and around 34.3% received doses above 60 Gy (gray). A study in Germany observed that patients receiving high radiation doses under RT had worse nutritional symptoms and a more notably weakened nutritional status, establishing that doses of ≥40 Gy in radiotherapy are predictive of the development of malnutrition by the treatment’s end [19]. In our study, only one patient received treatment with 30 Gy, while 98.5% underwent doses ≥ 40 Gy.

Among the main symptoms reported, dysphagia was observed in 34 patients in the first assessment, and around 65.9% (*p* = 0.002) of these were undernourished according to SGA. In the second assessment, 27 patients showed dysphagia, and 81.5% (*p* = 0.005) showed a prognostic score of 3, while 75.8% (*p* = 0.003) were undernourished according to SGA and 76.7% (*p* = 0.008) had a PNI ≤ 45.56.

A study on dysphagia emerging during radiotherapy sessions in China observed a 72.2% increase in the rate of manifestation of dysfunctions during treatment, with swallowing difficulties being related to weight loss, malnutrition according to SGA, the stage of disease, high doses of RT, concomitant treatments and cancer in the pharyngeal region [41]. Another study in Italy identified dysphagia in HN cancer patients in 48% of cases, while 37.95% developed it throughout the treatment, as well as showing greater weight loss, malnutrition, and a reduction in albumin and white blood cell levels. The study also reported that after one year of treatment, 58.62% died, and 41.38% of the patients were suffering from severe malnutrition [42].

Dysphagic patients consequently showed alterations in food consistency, as well as reduced diet volumes and the use of alternative feeding routes. In the first assessment, our study observed that the use of alternative feeding routes was more frequent in individuals suffering nutritional impacts (Score 1, 2 and 3), representing 72.7% of patients. In the second assessment, we ascertained that patients with a prognostic score of 3 had a significantly higher level of occurrence of alterations in ingested food volume (71%, *p* = 0.028), and undernourished patients (according to SGA (81.8%, *p* = 0.024) and PNI ≤ 45.56 (83.3%, *p* = 0.026) showed greater alterations in food volume. Besides this, patients with reduced PNI scores also showed a tendency towards alternative feeding routes (33.3%, *p* = 0.022), while patients with a score of 3 suffered from higher percentages of occurrence (90%) of this outcome.

An Asian study, conducted with a higher number of patients and over a longer period, identified as early as the pre-treatment stage that those with lower PNI scores were more susceptible to requiring alternative feeding routes [43]. A European study identified that 89.2% of dysphagic and undernourished patients were unable to use the oral route for nutrition [39]. Studies have reported that 1 in every 5 of such patients requires enteral feeding or ostomies, and when this is not an option, consistency alterations are necessary [40].

As previously mentioned, another predominant effect that we found was a reduction in the volume of ingested food, which could result in a nutritional imbalance, since the total caloric value ingested was also limited, and was not sufficient to meet the needs of exacerbated energy expenditure. All of this favors weight loss and the development of malnutrition, both of which are known to be markers of greater toxicity, worse responses to treatment and lower survival [18,19,20,44,45]. Furthermore, there is an increase in weight loss and a greater nutritional deficit when the cancer is located in the gastrointestinal region, mainly due to alterations in the digestive, metabolic and absorptive processes [46,47].

We observed that 22.38% of patients exhibited > 10% weight loss. An Italian study reported a percentage of patients with severe weight loss of 15.1%, associating the risk primarily with reduced PNI, and correlating it to greater negative impacts on patient survival [9]. A study carried out in Finland assessed the weight loss of patients undergoing cancer treatment, and found that 24% had severe weight loss before treatment, which figure rose to 70.7% at the end of the treatment [48].

Another factor presented in the literature as favoring weight loss and malnutrition is the stage of the tumor. Studies have shown that more advanced tumor stages have a greater nutritional impact [36], which is associated with reduced PNI values, as seen in studies on patients with colorectal and cervical cancer [49,50], and it is also associated with a diagnosis of malnutrition using SGA, as observed in a study carried out in Taiwan [51]. This study found that patients with a prognostic score of 2 suffered from a higher rate of occurrence of stage 3 cancer (31.6%, *p* = 0.018), with higher percentages of stage 3 (79%) and stage 4 (68.9%) cancer among those suffering a higher nutritional impact (Scores 1, 2, and 3).

Considering such interrelations, nutritionally weakened patients show greater risks of treatment interruption and prolonged hospital stay. Our data show that, in the first assessment, patients at nutritional risk (Score 1, 2, and 3) suffered higher percentages of treatment interruption (82.7%). In contrast, in the second assessment, we saw significantly more interruptions (of more than three days) of radiotherapy in patients with Score 3 (83.3%, *p* = 0.044), as well as in patients with PNI ≤ 45.56 (53.3%, *p* = 0.030). Similarly, a study carried out on HN patients in Taiwan found that malnourished patients with a low PNI received higher doses of medication and radiation, and they presented lower tolerance to treatment, a lower rate of therapy completion, as well as higher rates of toxicity and death [43].

The assessments using SGA, despite showing differing results, revealed a lower occurrence of interruption in well-nourished patients (11.1%). On the other hand, an Asian study on HN patients identified lower rates of treatment conclusion in undernourished patients identified using SGA [51]. Given these variations in findings, we stress the use of a combination of markers to improve sensitivity, ensuring better care coverage.

Concerning hospital admission, despite non-significance, we observed via the first and second assessments that Score 3 patients had higher rates of hospital admission (83.3%). In contrast, isolated markers showed lower percentages of medical care when patients had an improved nutritional status (SGA-A and PNI > 45.56). Studies on lung cancer patients assessed by SGA identified a requirement for longer hospital stays in undernourished patients [52]. Furthermore, another study on the characteristics of HN patients with unanticipated hospital admissions throughout treatment identified an increased risk for severely ill elderly people using alternative feeding routes, identifying the primary reasons as dehydration and gastrointestinal symptoms [53].

Cancer patient survival is strongly influenced by nutritional, immunological and inflammatory conditions, and malnutrition can lead to death in patients undergoing radiotherapy treatment in up to 20% of cases [29]. Although only 3% of the sample showed this outcome, around 79.1% (53 patients) were discharged with some complications having emerged during treatment that, if sustained over a long period, could lead to worse outcomes [6]. Based on the results presented, the second evaluation showed a tendency in patients with prognostic scores of 3 towards being discharged with more complications (92.3%, *p* = 0.059), and there were significantly more complications in patients with PNI ≤ 45.56 (40%, *p* = 0.029). Although this study did not assess mortality, studies carried out on cancer patients using PNI or SGA identified a shorter overall survival time and higher mortality when the values indicated malnutrition [9,12,30,33,43,54,55,56,57,58,59].

In the analyses carried out, the SGA has been proven to be a strong instrument with sensitivity when used in identifying toxicity, and especially malnutrition, from the first application, which may be related to its focus on symptoms of nutritional impact, and its use of anthropometric, dietary, and semiological assessments [60]. The PNI, on the other hand, showed more significant rates of toxicity after re-evaluation, and the nature of the markers used in its formula must be considered [9,37].

Albumin is a negative acute-phase protein with a long half-life, considered an indicator of nutritional status and inflammatory activity. At the beginning of the disease’s development, compensatory synthesis can be observed, but as stress and malnutrition worsen, its synthesis is suppressed [61]. Lymphocytes, on the other hand, are immune system cells that inhibit the proliferation of tumor cells, and their concentration is affected by the stress response, as well as deficient production due to malnutrition and reduction caused by factors such as radiosensitivity. Thus, in the early stages of the disease and in previously healthy patients, the values will be normalized, after which there will be a gradual and progressive reduction, making monitoring and reassessment important, as well as the association of markers as presented by the prognostic score [62,63,64].

The study had some limiting factors, such as the fact that It was carried out retrospectively and in a single center, which was recently inaugurated and is not yet operating at full capacity. As a result, we saw a reflection in the number of samples obtained, despite the one-year collection period. In addition, considering the use of data from medical records, it is possible that there may have been gaps or missing information, making it impossible to provide much detail on symptom levels and other information on adjuvant treatments.

However, considering the data presented, the nutritional markers offered by the prognostic score can be taken as an innovative tool for identifying malnutrition and toxicity, since they complement the limitations of the two indices (SGA and PNI). Their use enables the better monitoring of the individual’s prognosis, helping to devise a detailed clinical and nutritional therapeutic plan, especially after re-evaluating the scores, leading to a better quality of life, treatment completion rate, tolerability, and survival.

## 5. Conclusions

The use of prognostic scores has shown applicability in identifying weakened nutritional status and toxicities during cancer treatment, mainly after the second assessment, and demonstrated sufficient sensitivity to indicate reduced ingested food volume, treatment interruption, and dysphagia. Its use can also contribute to better nutritional planning and monitoring. The Subjective Global Assessment alone was sensitive to the identification of malnutrition and its correlation with the manifestation of dysphagia, xerostomia, and a reduction in ingested food volume. PNI alone, after the second assessment, was sensitive to the identification of alterations in ingested food volume, the use of alternative feeding routes, treatment interruption, dysphagia, and outcomes with three or more complications. Our data corroborate the published findings regarding undernourished patients with many complications. Despite our conclusions, we suggest further studies be undertaken with more patients and a greater monitoring time to derive more robust evidence.

## Figures and Tables

**Figure 1 nutrients-16-01363-f001:**
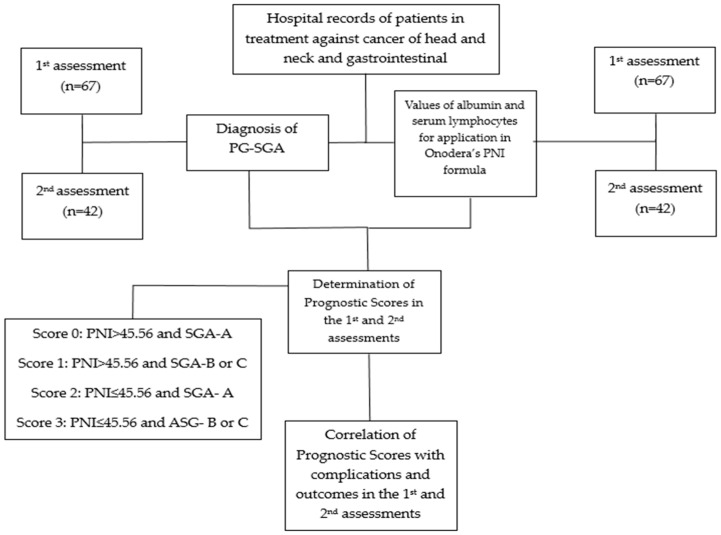
Fluxogram of methods utilized in the study. Abbreviations: PG-SGA (Patient-Generated Subjective Global Assessment); PNI (Prognostic Nutritional Index).

**Figure 2 nutrients-16-01363-f002:**
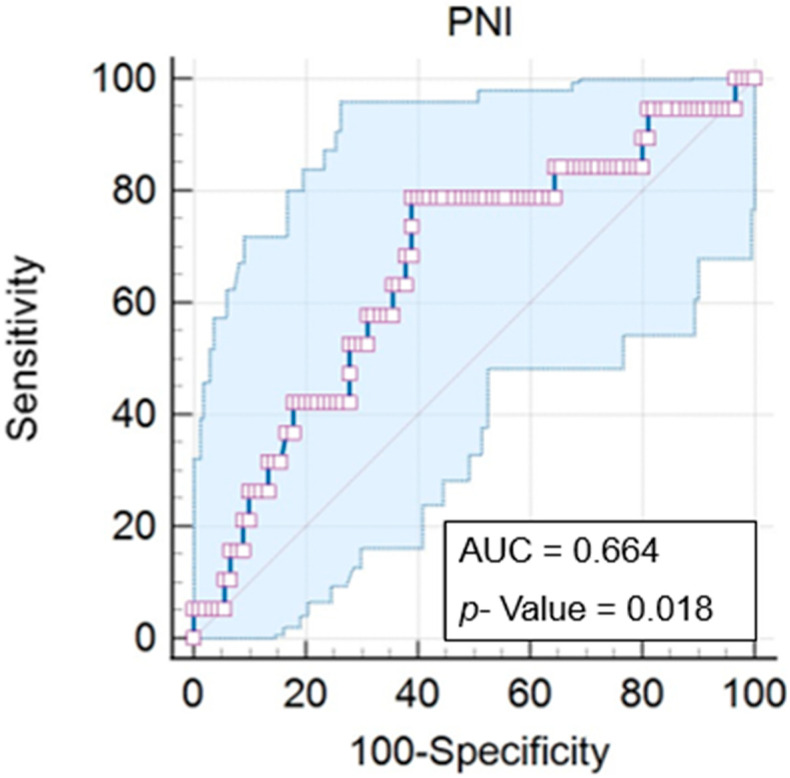
Analysis of the ROC curve for the Prognostic Nutritional Index (PNI).

**Figure 3 nutrients-16-01363-f003:**
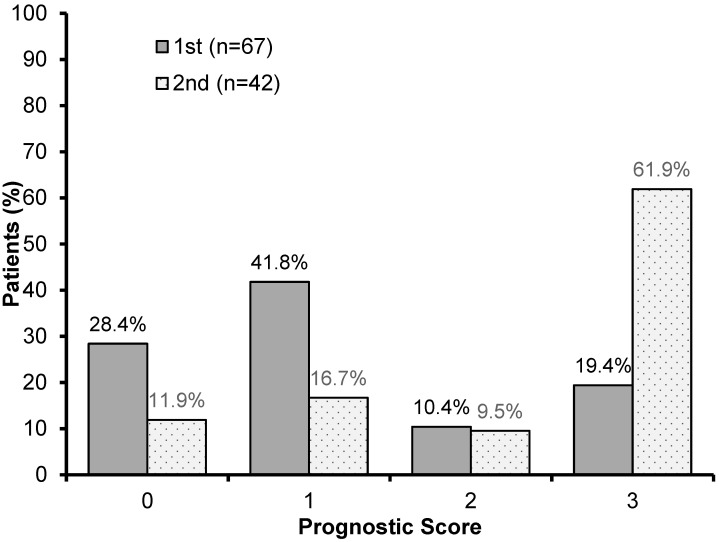
Frequency of distribution of the prognostic scores in the first and second assessments of patients.

**Figure 4 nutrients-16-01363-f004:**
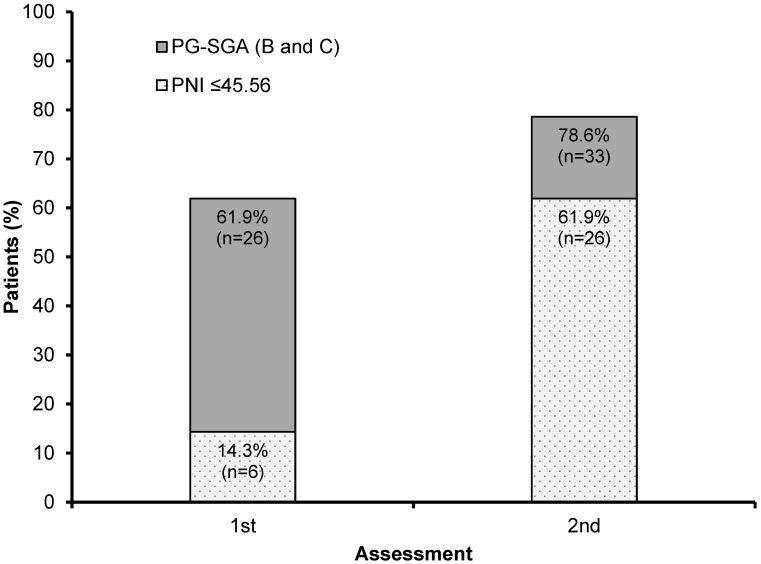
Relation of the percentages of undernourished patients identified using PG-SGA (B and C) with PNI ≤ 45.56 in the first and second assessments of the group subjected to two assessments (n = 42). Abbreviations—PG-SGA: Patient-Generated Subjective Global Assessment. PNI: Prognostic Nutritional Index.

**Table 1 nutrients-16-01363-t001:** Description of the prognostic score based on PNI and SGA-PPP.

Prognostic Score	PNI	PG-SGA
0	>45.56	A—Well-nourished
1	>45.56	B—Moderately undernourished or C—Severely malnourished
2	≤45.56	A—Well-nourished
3	≤45.56	B—Moderately undernourished or C—Severely Malnourished

Source: Author. PNI: Prognostic Nutritional Index. PG-SGA: Patient-Generated Subjective Global Assessment.

**Table 2 nutrients-16-01363-t002:** General data and clinical characteristics.

Variables	% (n)
TOTAL	100 (67)
GENDER	
Female	25.4 (17)
Male	74.6 (50)
AGE	
≤59 years	40.3 (27)
≥60 years	59.7 (40)
CANCER LOCATION	
Tongue and oral cavity	2.99 (2)
Pharynx	28.35 (19)
Larynx	8.95 (6)
Esophagus	17.91 (12)
Parotid and salivary glands	4.47 (3)
Nasal cavity and facial sinuses	4.47 (3)
Stomach	1.5 (1)
Rectum	23.9 (16)
Anal canal	7.46 (5)
NUMBER OF SESSIONS	
≤20 sessions	11.9 (8)
21 to 29 sessions	47.8 (32)
≥30 sessions	40.3 (27)
PNI	
≤45.56	28.4 (19)
>45.56	71.6 (48)
PG-SGA	
A—Well-nourished	38.8 (26)
B—Moderately undernourished	46.3 (31)
C—Severely malnourished	14.9 (10)
OUTCOME	
Hospital Discharge	97 (65)
Death	3 (2)
CHEMOTHERAPY	
Yes	74.6 (50)
No	25.4 (17)
PROGNOSTIC SCORE	
Score 0 (PNI > 45.56 and SGA-A)	28.4 (19)
Score 1 (PNI > 45.56 and SGA-B or C)	41.8 (28)
Score 2 (PNI ≤ 45.56 and SGA-A)	10.4 (7)
Score 3 (PNI ≤ 45.56 and SGA-B or C)	19.4 (13)

PNI: Prognostic Nutritional Index. PG-SGA: Patient-Generated Subjective Global Assessment.

**Table 3 nutrients-16-01363-t003:** Description of variables and outcomes in relation to the classification of Prognostic Scores for patients.

Variables	Total Patients (n = 67)	Score 0 (n = 19/28.4%)	Score 1 (n = 28/41.8%)	Score 2 (n = 7/10.4%)	Score 3 (n = 13/19.4%)	*p*-Value
STAGES						
I	2	0 (0) ^a^	100 (2) ^a^	0 (0) ^a,b^	0 (0) ^a^	0.018
II	17	35.3 (6) ^a^	52.9 (9) ^a^	5.9 (1) ^a,b^	5.9 (1) ^a^	
III	19	21.1 (4) ^a^	21.1 (4) ^a^	31.6 (6) ^b^	26.3 (5) ^a^	
IV	29	31 (9) ^a^	44.8 (13) ^a^	0 (0) ^a^	24.1 (7) ^a^	
DOSE RT (Gy)						
≤60	44	31.8 (14)	36.4 (16)	15.9 (7)	15.9 (7)	0.116
≥60.1	23	21.7 (5)	52.2 (12)	0 (0)	26.1 (6)	
NUMBER OF SESSIONS						
≤20 sessions	8	25 (2)	25 (2)	0 (0)	50 (4)	0.108
21 to 29 sessions	32	28.1 (9)	43.75 (14)	18.8 (6)	9.4 (3)	
≥30 sessions	27	29.6 (8)	44.4 (12)	3.7 (1)	22.2 (6)	
CONCOMITANT CHEMOTHERAPY						
Yes	50	22 (11)	45 (23)	14 (7)	18 (9)	0.101
No	17	47.1 (8)	29.4 (5)	0 (0)	23.5 (4)	
WEIGHT LOSS						
0 TO 5.00%	37	24.3(9)	40.5 (15)	13.5 (5)	21.6 (8)	0.342
5.01 TO 10%	15	46.7 (7)	26.7 (4)	13.3 (2)	13.3 (2)	
>10.1%	15	20 (3)	60 (9)	0 (0)	20 (3)	
OUTCOME						
without complications	12	33.3 (4)	41.7 (5)	8.3 (1)	16.7 (2)	0.588
with <3 complications	37	29.7 (11)	35.1 (13)	16.2 (6)	21.6 (8)	
≥3 complications	18	27.8 (5)	55.5 (10)	0 (0)	16.7 (3)	

RT: radiotherapy. Gy: grays. Values with different letters in the same line indicate significant differences between the groups (*p* < 0.05).

**Table 4 nutrients-16-01363-t004:** Relation between complications and the prognostic scores in the first assessment of patients.

Variables	Total Patients (n = 67)	Score 0 (n = 19/28.4%)	Score 1 (n = 28/41.8%)	Score 2 (n = 7/10.4%)	Score 3 (n = 13/19.4%)	*p*-Value
Alteration in consistency						
YES	16	43.8 (7)	31.3 (5)	0 (0)	25 (4)	0.183
NO	51	23.5 (12)	45.1 (23)	13.7 (7)	17.6 (9)	
Alteration in volume						
YES	45	31.1 (14)	46.7 (21)	4.4 (2)	17.8 (8)	0.109
NO	22	22.7 (5)	31.8 (7)	22.7 (5)	22.7 (5)	
Alternative feeding route						
YES	11	27.3 (3)	54.5 (6)	0 (0)	18.2 (2)	0.593
NO	56	28.6 (16)	39.3 (22)	12.5 (7)	19.6 (11)	
Hospital admission						
YES	12	16.7 (2)	50 (6)	8.3 (1)	25 (3)	0.744
NO	55	30.9 (17)	40 (22)	10.9 (6)	18.2 (10)	
Treatment interruption						
YES	29	17.2 (5)	44.8 (13)	17.2 (5)	20.7 (6)	0.200
NO	38	36.8 (14)	39.5(15)	5.3 (2)	18.4 (8)	
Mucositis						
YES	29	31(9)	41.4 (12)	0 (0)	27.6 (8)	0.065
NO	38	26.3 (10)	42.1 (16)	18.4 (7)	13.2 (5)	
Dermatitis						
YES	49	26.5 (13)	49 (25)	8.2 (4)	16.3 (8)	0.234
NO	18	33.3 (6)	22.2 (4)	16.7 (3)	27.8 (5)	
Dysphagia						
YES	34	20.6 (7) ^a^	52.9 (18) ^a^	0 (0) ^b^	26.5 (9) ^a^	0.006
NO	33	36.4 (12) ^a^	30.3 (10) ^a^	21.2 (7) ^a^	12.1 (4) ^a^	
Odynophagia						
YES	5	0 (0)	40 (2)	0 (0)	60 (3)	0.082
NO	62	30.6 (19)	41.9 (26)	11.3 (7)	16.1 (10)	
Xerostomia						
YES	13	46.15 (6)	46.15 (6)	0 (0)	7.7 (1)	0.195
NO	54	24.1 (13)	40.7 (22)	13 (7)	22.2 (12)	
Diarrhea						
YES	11	36.3 (4)	36.3 (4)	27.3 (3)	0 (0)	0.080
NO	56	26.8 (15)	42.9 (24)	7.1 (4)	23.2 (13)	

Values with different letters in the same line indicate significant difference between the groups (*p* < 0.05).

**Table 5 nutrients-16-01363-t005:** Relation between complications and PG-SGA and PNI scores of patients in the first assessment.

Variables	Total (n = 67)	SGA-A (n = 26/38.8%)	SGA-B and C (n = 41/61.2%)	*p*-Value	PNI ≤ 45.56 (n = 19/28.4%)	PNI > 45.56 (n = 48/71.6%)	*p*-Value
Alteration of consistency				0.642			0.328
YES	16	26.9 (7)	22 (9)		15.8 (3)	27.1 (13)	
NO	51	73.1 (19)	78 (32)		84.2 (16)	72.9 (35)	
Alteration in volume				0.435			0.030
YES	45	61.5 (16)	70.7 (29)		47.4 (9) ^a^	75 (36) ^b^	
NÃO	22	38.5 (10)	29.3 (12)		52.6 (10) ^a^	25 (12) ^b^	
Alternative feeding route				0.391			0.413
YES	11	11.5 (3)	19.5 (8)		10.5 (2)	18.8 (9)	
NO	56	88.5 (23)	80.5 (33)		89.5 (17)	81.2 (39)	
Hospital admission				0.279			0.673
YES	12	11.5 (3)	22 (9)		21.1 (4)	16.7 (8)	
NO	55	88.5 (23)	78 (32)		78.9 (15)	83.3 (40)	
Treatment interruption				0.526			0.129
YES	29	38.5 (10)	46.3 (19)		57.9 (11)	37.5 (18)	
NO	38	61.5 (16)	53.7 (22)		42.1 (8)	62.5 (30)	
Mucositis				0.254			0.503
YES	29	34.6 (9)	48.8 (20)		36.8 (7)	45.8 (22)	
NO	38	65.4 (17)	51.2 (21)		63.2 (12)	54.2 (26)	
Dermatitis				0.254			0.077
YES	49	65.4 (17)	78 (32)		57.9 (11)	79.2 (38)	
NO	18	34.6 (9)	22 (9)		42.1 (8)	20.8 (10)	
Dysphagia				0.002			0.373
YES	34	26.9 (7) ^a^	65.9 (27) ^b^		42.1 (8)	54.2 (26)	
NO	33	73.1 (19) ^a^	34.1 (14) ^b^		57.9 (11)	45.8 (22)	
Odynophagia				0.064			0.103
SIM	5	0 (0)	12.2 (5)		15.8 (3)	4.2(2)	
NO	62	100 (26)	87.8 (36)		84.2 (16)	95.8 (46)	
Xerostomia				0.542			0.066
YES	13	23.1 (6)	17.1 (7)		5.3 (1)	25 (12)	
NO	54	76.9 (20)	82.9 (34)		94.7 (18)	75 (36)	
Diarrhea				0.065			0.930
YES	11	26.9 (7)	9.8 (4)		15.8 (3)	16.7 (8)	
NO	56	73.1 (19)	90.2 (37)		84.2 (16)	83.3 (40)	
OUTCOME							0.347
Without complications	12	19.2 (5)	17.1 (7)	0.530	15.8 (3)	18.8 (9)	
With <3 complications	37	61.5 (16)	51.2 (21)		68.4 (13)	50 (24)	
With ≥3 complications	18	19.2 (5)	31.7 (13)		15.8 (3)	31.3 (15)	

SGA: Subjective Global Assessment. PNI: Prognostic Nutritional Index. Values with different letters in the same line indicate significant difference between the groups (*p* < 0.05).

**Table 6 nutrients-16-01363-t006:** Relations between complications associated with the second prognostic score in patients in the second assessment.

Variables	Total of Patients (n = 42)	Score 0 (n = 5/11.9%)	Score 1 (n = 7/16.7%)	Score 2 (n = 4/9.5%)	Score 3 (n = 26/61.9%)	*p*-Value
Consistence alteration						
YES	13	7.7 (1)	23.1 (3)	0 (0)	69.2 (9)	0.440
NO	29	13.8 (4)	13.8 (4)	13.8 (4)	58.6 (17)	
Volume alteration						
YES	31	3.2 (1) ^b^	16.1 (5)	9.7 (3)	71 (22) ^a^	0.028
NO	11	36.3 (4) ^a^	18.2 (2)	9.1 (1)	36.3 (4) ^b^	
Alternative feeding route						
YES	10	0 (0)	0 (0)	10 (1)	90 (9)	0.143
NO	32	15.6 (5)	21.9 (7)	9.4 (3)	53.1 (17)	
Hospital admission						
YES	6	16.7 (1)	0 (0)	0 (0)	83.3 (5)	0.478
NO	36	11.1 (4)	19.4 (7)	11.1 (4)	58.3 (21)	
Treatment interruption						
YES	18	11.1 (2)	0 (0) ^b^	5.6 (1)	83.3 (15) ^b^	0.044
NO	24	12.5 (3)	29.2 (7) ^a^	12.5 (3)	45.8 (11) ^a^	
Mucositis						
YES	23	13 (3)	13 (3)	4.3 (1)	69.6 (16)	0.500
NO	19	10.5 (2)	21.1 (4)	15.8 (3)	52.6 (10)	
Dermatitis						
YES	33	6.1 (2)	21.2 (7)	9.1 (3)	63.6 (21)	0.092
NO	9	33.3 (3)	0 (0)	11.1 (1)	55.6 (5)	
Dysphagia						
YES	27	3.7 (1) ^b^	11,1(3)	3.7 (1)	81.5 (22) ^b^	0.005
NO	15	26.7 (4) ^a^	26.7 (4)	20 (3)	26.7 (4) ^a^	
Odynophagia						
YES	2	0 (0)	0 (0)	0 (0)	100 (2)	0.731
NO	40	12.5 (5)	17.5 (7)	10 (4)	60 (24)	
Xerostomia						
YES	11	0 (0)	18.2 (2)	0 (0)	81.8 (9)	0.244
NO	31	16.1 (5)	16.1 (5)	12.9 (4)	54.8 (17)	
Diarrhea						
YES	5	20 (1)	20 (1)	0 (0)	60 (3)	0.827
NO	37	10.8 (4)	16.2 (6)	10.8 (4)	62.2 (23)	
OUTCOME						
Without complications	6	33.3 (2)	33.3 (2)	16.7 (1)	16.7 (1)	0.059
With <3 complications	23	8.7 (2)	21.7 (5)	13 (3)	56.4 (13)	
With ≥3 complications	13	7.7 (1)	0 (0)	0 (0)	92.3 (12)	

Values with different letters in the same line indicate significant difference between the groups (*p* < 0.05).

**Table 7 nutrients-16-01363-t007:** Relation between complications and PG-SGA and PNI scores of patients in the second assessment.

Variables	Total (n = 42)	SGA-A (n = 9/21.4%)	SGA-B and C (n = 33/78.6%)	*p*-Value	PNI ≤ 45.56 (n = 30/71.4%)	PNI > 45.56 (n = 12/28.6%)	*p*-Value
Alteration in consistency							
YES	13	11.1 (1)	36.4 (12)	0.146	30 (9)	33.3 (4)	0.833
NO	29	88.9 (8)	63.6 (21)		70 (21)	66.7 (8)	
Alteration in volume							
YES	31	44.4 (4) ^a^	81.8 (27) ^b^	0.024	83.3 (25) ^a^	50 (6) ^b^	0.026
NO	11	55.6 (5) ^a^	18.2 (6) ^b^		16.7 (5) ^a^	50 (6) ^b^	
Alternative feeding route							
YES	10	11.1 (1)	27.3 (9)	0.313	33.3 (10) ^a^	0 (0) ^b^	0.022
NO	32	88.9 (8)	72.7 (24)		66.7 (20) ^a^	100 (12) ^b^	
Hospital admission							
YES	6	11.1 (1)	15.2 (5)	0.759	16.7 (5)	8.3 (1)	0.486
NO	36	88.9 (8)	84.8 (28)		83.3 (25)	91.7 (11)	
Treatment interruption							
YES	18	33.3 (3)	45.5 (15)	0.515	53.3 (16) ^a^	16.7 (2) ^b^	0.030
NO	24	66.7 (6)	51.5 (18)		46.7 (14) ^a^	83.3 (10) ^b^	
Mucositis							
YES	23	44.4 (4)	57.6 (19)	0.483	56.7 (17)	50 (6)	0.695
NO	19	55.6 (5)	42.4 (14)		43.3 (13)	50 (6)	
Dermatitis							
YES	33	55.6 (5)	84.8 (28)	0.058	80 (24)	75 (9)	0.721
NO	9	44.4 (4)	15.2 (5)		20 (6)	25 (3)	
Dysphagia							
YES	27	22.2 (2) ^a^	75.8 (25) ^b^	0.003	76.7 (23) ^a^	33.3 (4) ^b^	0.008
NO	15	77.8 (7) ^a^	24.2 (8) ^b^		23.3 (7) ^a^	66.7 (8) ^b^	
Odynophagia							
YES	2	0 (0)	6.1 (2)	0.449	6.7 (2)	0 (0)	0.359
NO	40	100 (9)	93.9 (31)		93.3 (28)	100 (12)	
Xerostomia							
YES	11	0 (0) ^a^	33.3 (11) ^b^	0.044	30 (9)	16.7 (2)	0.375
NO	31	100 (9) ^a^	66.7 (22) ^b^		70 (21)	83.3 (10)	
Diarrhea							
YES	5	11.1 (1)	12.1 (4)	0.934	10 (3)	16.7 (2)	0.547
NO	37	88.9 (8)	87.9 (29)		90 (27)	83.3 (10)	
Outcome							
without complications	6	33.3 (3)	9.1 (3)	0.530	6.7 (2) ^a^	33.3 (4) ^b^	0.029
With <3 complications	23	55.6 (5)	54.5 (18)		53.3 (16) ^a^	58.3 (7) ^a^	
≥3 complications	13	11.1 (1)	36.4 (12)		40 (12) ^a^	8.3 (1) ^b^	

SGA: Subjective Global assessment. PNI: Prognostic Nutritional Index. Values with different letters in the same line indicate significant difference between the groups (*p* < 0.05).

## Data Availability

Data is contained within the article.

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
