# Peer review of "Nutritional Prognosis of Patients Submitted to Radiotherapy and Its Implications in Treatment"

_nutrients, 2024, doi:10.3390/nu16091363_

Round 1

Reviewer 1 Report

Comments and Suggestions for Authors

I read with interest the article written by Molina Irigaray et al. The study analysed the nutritional aspects of patients with gastrointestinal and head and neck cancer.  Patients with cancer often have compromised nutritional status due to increased energy requirements and inflammatory responses. Gastrointestinal and head and neck cancers can further compromise nutritional status due to difficulties with mastication, swallowing, digestion, and nutrient absorption. Radiotherapy may cause side effects, including nausea, vomiting, diarrhea, odynophagia, and dysphagia. These side effects can lead to a decrease in energy intake and worsen the nutritional status of patients.
Previous studies have evaluated the nutritional status and prognosis of pre-surgical patients. However, there is a lack of research on patients undergoing non-surgical treatments, such as radiotherapy. Analysing the nutritional intake of patients undergoing radiotherapy can help identify nutritional risks and anticipate complications. This allows for early intervention and better clinical and nutritional recovery.

This retrospective cross-sectional study analysed secondary data from hospital records of patients who underwent radiotherapy between July 2022 and July 2023. Prognostic Scores were established using the Prognostic Nutritional Index (PNI) and Subjective Global Assessment (SGA), which were assessed at the beginning and end of treatment. Patients were categorised based on their PNI and SGA values.  The study assessed symptoms such as odynophagia, reduced feeding intake, dysphagia, altered feeding frequency, xerostomia and the need for alternative feeding strategies. The study also evaluated hospital admission rates and the impact of nutritional symptoms on patients' nutritional status. The Prognostic Scores, which combine the Prognostic Nutritional Index (PNI) and Subjective Global Assessment (SGA), were found to be sensitive indicators of complications and outcomes. Patients with a PNI of 45.56 or lower and malnutrition based on SGA had a higher incidence of odynophagia, reduced diet volume, dysphagia, treatment interruption, and the need for alternative feeding methods. SGA alone was sensitive to changes in diet volume, dysphagia, and xerostomia in the second assessment. Conversely, well-nourished patients experienced fewer dietary changes, required fewer alternative feeding strategies, had lower hospital admission rates and experienced fewer treatment interruptions.

The article reports on a single institution cohort of gastrointestinal cancers and head and neck cancers analysed together.
-It would be interesting to differentiate the two case scenarios, also because the nutritional scenario and implication are different for head and neck and oesophageal versus stomach versus rectal cancer.

-Additionally, evaluating the scores analysed in the two patient situations would be useful.
-The text lacks information on operated patients or those treated in adjuvant, and does not differentiate between types of chemotherapy used.

-The radiotherapy doses are also unclear, and the cut-off of 20 sessions is debatable and should be read in relation to primary and fractionation.

-What is the meaning of treatment interruptions? How many days were deemed to be significant?
- In Table 4, all parameters (interruptions, mucositis, dysphagia, etc.) were scored as "yes/no". It might be useful to include cut-off values based on reference scales in the table.

-Overall, as a rationale and approach, I find the article very interesting from a medical perspective. As mentioned, it fills a gap in the literature.
-A more comprehensive case definition (perhaps with separate analyses) and  more data on radio- and chemotherapy treatments would be useful.
-Additionally, a revision of the English writing is suggested.

Comments on the Quality of English Language

Revision required

Author Response

Campo Grande, february, 29, 2024

Dear Reviewer 1,

We are submitting the manuscript titled “Nutritional Prognosis of Patients Submitted to Radiotherapy and Its Implications in Treatment” for possible publication in the Nutrients.

We thank you for your considerations in our manuscript with the purpose of improving our article. We analyzed and modified our article according your considerations and the changes are listed below. In the manuscript, these changes are highlighted in pink.

- The article reports on a single institution cohort of gastrointestinal cancers and head and neck cancers analysed together.
-It would be interesting to differentiate the two case scenarios, also because the nutritional scenario and implication are different for head and neck and oesophageal versus stomach versus rectal cancer.
-Additionally, evaluating the scores analysed in the two patient situations would be useful.

Response: Regarding the joint analysis of gastrointestinal and head and neck cancers, it was carried out considering the union of anatomical regions that bring greater complications in nutritional status given their functionalities. It is understood that there are different implications, such as a greater tendency to dysphagia in patients with head and neck cancer or a greater frequency of diarrhea in patients with colorectal cancer, but even so, there will be direct damage to patients' nutritional status in both cases. In addition, as mentioned in the note, the study was carried out with a single institutional cohort, which is equivalent to the state's reference service, but because it is a recently inaugurated service, it resulted in a reduced number of samples, so we opted to group the sites as "cancers with nutritional impact" for statistical feasibility.

-The text lacks information on operated patients or those treated in adjuvant, and does not differentiate between types of chemotherapy used.

Response: We appreciate the comment and agree that it could bring new information to the study. However, it should be clarified that the study was carried out retrospectively by analyzing patient records and, for the most part, no specific information was found on the type of chemotherapy or surgical procedures carried out among the patients taking part in the study, justifying the absence of this information. Furthermore, subdividing the sample into subgroups depending on other interventions would have made statistical interpretation more difficult.

-The radiotherapy doses are also unclear, and the cut-off of 20 sessions is debatable and should be read in relation to primary and fractionation.

Response: Thank you for your important comment. Regarding the radiation dose, knowing that the radiation received ranged from 30 to 70gy, with daily doses of 1.5 to 3gy/day 5 times a week, also considering the average received of 55.9gy and the dose values used in other studies (Diaz et al, 2021; Löser et al.,2022; Finger et al., 2023; Xu et al., 2024) a radiation dose cut-off point of 60gy was established. This information was included in the methodology, presented in lines 96-99. With regard to the number of sessions, a stratification was established based on the survey of the number of sessions carried out, which ranged from 10 to 35. In order to establish a classification range of a maximum of 3 indices, so as not to jeopardize the statistical analyses, it was decided to use 20 as the lower cut-off point, considering that only 8 patients had undergone this number of sessions. This information was included in the methodology, presented in lines 93-99.

What is the meaning of treatment interruptions? How many days were deemed to be significant?

Response: Treatment interruptions were considered to be breaks in radiotherapy due to symptoms triggered during the sessions, such as mucositis, dysphagia, odynophagia, dermatitis and diarrhea, which are shown in tables 4, 5, 6 and 7. A minimum of 3 days' break was considered relevant, according to the profile of the sample and findings in the literature that reinforced the clinical and prognostic complications after 3 days of interruption.  We thank you for your comment and inform you that the adjustment was made to the methodology in lines 118-119.

- In Table 4, all parameters (interruptions, mucositis, dysphagia, etc.) were scored as "yes/no". It might be useful to include cut-off values based on reference scales in the table.

Response: Regarding the inclusion of a reference in the parameters of the alterations found in the patients (mucositis, dysphagia, odynophagia, diarrhea), considering that the study was carried out retrospectively using data from the medical records of a public radiotherapy outpatient clinic in Brazil, not all the symptoms described had the severity levels according to the CTC. Furthermore, not all of the alterations presented had levels of manifestation, which would have compromised the analysis and presentation. We understand and thank you for your question.

-A more comprehensive case definition (perhaps with separate analyses) and  more data on radio- and chemotherapy treatments would be useful.

Response: We would like to thank you for your comments and reflections. We would like to inform you that the target audience, gastrointestinal and head and neck cancer, was chosen considering the tendency for greater nutritional impacts according to the data in the literature raised in the article. The radiotherapy outpatient clinic used is a public reference service in the region, but it was recently inaugurated, so there was a reflection in the sample number, making separate analyses impossible. However, this stimulates the possibility of more comprehensive studies to be carried out at the unit over a longer period of time, encouraging health research.

-Additionally, a revision of the English writing is suggested.

Response: We would like to inform you that the article has been submitted to the English Language Editing Services - MDPI, according to the certificate attached.

REFERENCES

Diaz, C.; Hayward, C. J.; Safoine, M.; Paquette, C.; Langevin, J.; Galarneau, J.; Théberge, V.; Ruel, J.; Archambault, L.; Fradette, J. Ionizing Radiation Mediates Dose Dependent Effects Affecting the Healing Kinetics of Wounds Created on Acute and Late Irradiated Skin. Surgeries 2021, 2, 35-37.

Löser, A.; Grohmann, M.; Dedo, A.; Greinert, F.; Krause, L.; Molwitx, I.; Krull, A.; Petersen, C. Impact of dosimetric factors on long-term percutaneous enteral gastrostomy (PEG) tube dependence in head and neck cancer patients after (chemo)radiotherapy—results from a prospective randomized trial. Strahlenther Onkol. 2022, 198, 1016-1024.

Finger, A.; Grohmann, M.; Krause, L.; Krüll, A.; Petersen, C.; Thieme, A.; Rades, D.; Löser, A. Irradiation dose to the swallowing apparatus impacts nutritional status in head and neck cancer patients—results from the prospective randomized HEADNUT trial. Strahlenther Onkol. 2023, 199, 875-880.

Xu, T.; Shen, C.; Zhou, X.; Zhu, L.; Xiang, J.; Wang, Y.; Zhu, Y.; He, X.; Ying, H.; Whang, Y.; Ji, Q.; Hu, C.; Lu, X. Selective Treatment Deintensification by Reducing Radiation Dose and Omitting Concurrent Chemotherapy Based on Response to Induction Chemotherapy in Human Papillomavirus-Associated Oropharyngeal Squamous Cell Carcinoma: A Single-Arm, Phase 2 Trial (IChoice-01). Int. J. Radiat. Oncol. Biol. Phys. 2024, 117, 169-178.

Yao, JJ.; Jin, YN.; Wang, SJ.; Zhang, F.; Zhou, GQ.; Zhang, WJ.; Cheng, ZB.; Ma J.; Qi,ZY.; Sun, Y. The detrimental effects of radiotherapy interruption on local control after concurrent chemoradiotherapy for advanced T-stage nasopharyngeal carcinoma: an observational, prospective analysis. BMC Cancer 2018, 18, 740.

Yang, XL; Zhou, GQ.; Lin, L.; Zhang, LL.; Zhen, FP.; Lv, JW.; Kou, J.; Wen, DW.; Ma, J.; Sun, Y.; Mao, YP. Prognostic value of radiation interruption in different periods for nasopharyngeal carcinoma patients in the intensity‐modulated radiation therapy era. Cancer Med. 2021, 10, 143-155.

Thank you for your considerations,                                                     

Best regards,

Mariana Maroso Molina Irigaray.

Reviewer 2 Report

Comments and Suggestions for Authors

This is an interestinmg research study with adequate quality and quite novelty. Some points should be sddressed.

- In the Introduction section, the authors should merge the first three paragraph into one.

- In the introduction section the authors should emphasize the role of malnutrition with cancer diagnosis and prognosis by added more recent references and relevant statement such as  

 doi: 10.3390/nu15245068

doi: 10.3390/medsci11040064

doi: 10.1007/s11596-023-2808-4

doi: 10.1080/01635581.2017.1367947

doi: 10.1007/s00520-022-07242-9

- At the end of Introduction section, the authors should emphasize the existing literature gap for which they performed their study with the aim to cover this gap.

- Concerning the astatistical analysis of the study data, why a multivariate logistic statistical analysis did not perform to establish which variables remain indepedent after adjusting for potential confounding factors?

- The Discussion section is quite large and it should be condensed in some point avoiding repetition with the introduction.

- At the end of the Discussion section, a paragraph with the strengths and the limitation of the study are strongly recommended.

Comments on the Quality of English Language

Moderate editing of English language required

Author Response

Campo Grande, february, 29, 2024

Dear Reviewer 2,

We are submitting the manuscript titled “Nutritional Prognosis of Patients Submitted to Radiotherapy and Its Implications in Treatment” for possible publication in the Nutrients.

We thank you for your considerations in our manuscript with the purpose of improving our article. We analyzed and modified our article according your considerations and the changes are listed below. In the manuscript, these changes are highlighted in green.

- In the Introduction section, the authors should merge the first three paragraph into one.

Response: We thank you for your comments and inform you that the adjustments have been made. As noted in lines 31-40, the information has been condensed into one paragraph to follow the same line of reasoning.

- In the introduction section the authors should emphasize the role of malnutrition with cancer diagnosis and prognosis by added more recent references and relevant statement such as   DOI: 10.3390/nu15245068; DOI: 10.3390/medsci11040064; DOI: 10.1007/s11596-023-2808-4; DOI: 10.1080/01635581.2017.1367947; DOI: 10.1007/s00520-022-07242-9

Response: We would like to inform you that the references suggested in the introduction and discussion have been included, according to the description of the lines in each DOI indicated. In addition, the impact of malnutrition on the patient's prognosis has been better explained, as indicated in lines 41-60.

DOI: 10.3390/nu15245068à lines 41-44

DOI: 10.3390/medsci11040064 à lines 44-49; 303-305

DOI: 10.1007/s11596-023-2808-4 à lines 56-60

DOI: 10.1080/01635581.2017.1367947à lines 50-51

DOI: 10.1007/s00520-022-07242-9 à lines 36-39; 52-56; 303-305

- At the end of Introduction section, the authors should emphasize the existing literature gap for which they performed their study with the aim to cover this gap.

Response: We would like to inform you that we have added to the gap in the literature regarding the lack of studies associating indices for assessing nutritional status and prognosis, as well as the lack of prognostic studies on patients undergoing radiotherapy treatments. These considerations can be seen in lines 56-60.

- Concerning the statistical analysis of the study data, why a multivariate logistic statistical analysis did not perform to establish which variables remain indepedent after adjusting for potential confounding factors?

Response: Thank you for your comment. We did not carry out a multivariate analysis since the main objective of the study was to analyze the prognosis of patients undergoing radiotherapy based on the PNI and PG-SGA markers in association, in order to relate poor nutritional status to a greater number of treatment complications, i.e. the objective of the study was to evaluate how the Prognosis/Nutritional Status Score would determine possible alterations and not the other way around. We should also point out that, in multivariate analysis, the outcome needs to be binomial, but the scores were classified into four groups, which would not allow such an analysis for this variable.

- The Discussion section is quite large and it should be condensed in some point avoiding repetition with the introduction.

Response: Thank you for your comment. We have revised the Discussion section and reduced the text as much as possible, avoiding compromising information that we believe is essential to understanding our results. We would like to clarify that a suggestion was made to us by another reviewer to include points regarding the albumin and lymphocyte markers in the PNI formula. The new information has slightly extended the length of the discussion.

- At the end of the Discussion section, a paragraph with the strengths and the limitation of the study are strongly recommended.

Response: We would like to inform you that the discussion includes the study's limiting and promising factors, as shown in lines 446-458. One of the limitations was the fact that the study was carried out retrospectively and using secondary data, depending on the information contained in the patients' medical records. In addition, the fact that only one radiotherapy treatment center was analyzed, which was recently inaugurated and still has a limited treatment capacity, resulted in a small sample size, despite the one-year collection period. On the other hand, the data obtained in the study reinforces the better prediction of toxicities with the associated use of indicators, which complement each other in their weaknesses, making it possible to anticipate nutritional approaches, ensuring a better quality of life, treatment development and survival.

P.S. We would like to inform you that the article has been submitted to the English Language Editing Services - MDPI, according to the certificate attached.

Thank you for your considerations,                                                     

Best regards,

Mariana Maroso Molina Irigaray.

Reviewer 3 Report

Comments and Suggestions for Authors

This prominent article shows the benefit of using Nutritional risk scores in patients undergoing radiotherapy. I have some remarks:

Methods: Chi Square Test: why did you you choose Bonferroni's correction over Yates' correction?

Results: Tables: The severity of the side-effects of chemo- and radiotherapy are usually classified according to the CTC (grade 1 to 4). Is any information concerning this available?

Discussion: The PNI is based on albumine measurement and lymphocyte count. Both are rather late markers, pointing out already severe malnutrition, and the authors should comment on this.

Comments on the Quality of English Language

A few typo's throughout the text.

Author Response

Campo Grande, february, 29, 2024

Dear Reviewer 3,

We are submitting the manuscript titled “Nutritional Prognosis of Patients Submitted to Radiotherapy and Its Implications in Treatment” for possible publication in the Nutrients.

We thank you for your considerations in our manuscript with the purpose of improving our article. We analyzed and modified our article according your considerations and the changes are listed below. In the manuscript, these changes are highlighted in blue.

Methods: Chi Square Test: why did you choose Bonferroni's correction over Yates' correction?

Response: Thank you for your comment. Fisher's Exact Test or Yates' Correction is used when analyzing small samples with only two variables (the so-called 2 by 2 tables), but in this study there were situations of simultaneous comparisons of more variables. Thus, considering the data presented, there was a need for a multiple comparisons test between proportions and, to this end, in the chi-square test analyses where there was statistical significance, the Bonferroni correction was carried out to determine the real significant indicator between the groups, making the test more demanding. This information has been added to the methodology section on lines 129-130.

Results: Tables: The severity of the side-effects of chemo- and radiotherapy are usually classified according to the CTC (grade 1 to 4). Is any information concerning this available?

Response: We understand and thank you for your question. We are aware of the classifications made by the CTC, but throughout the study, we chose not to present the classifications for each variable described, as this would have a significant impact on the statistical analysis. In addition, it is important to note that the study was carried out retrospectively using data from the medical records of a public radiotherapy outpatient clinic in Brazil, and not all the symptoms described had the severity levels according to the CTC.

Discussion: The PNI is based on albumine measurement and lymphocyte count. Both are rather late markers, pointing out already severe malnutrition, and the authors should comment on this.

Response:  Thank you for your comment. The use of the Onodera formula based on lymphocyte and albumin values has been widespread among prognostic studies of cancer patients, since they are also markers that are easy to apply, especially in public health services in our country. Considering that the scores were carried out at two points in time, at the beginning and end of treatment, with more promising data being seen after the second evaluation, the reflections pointed out in relation to the fact that the markers are late, as can be seen in lines 433-445, were included in the discussion.

P.S. We would like to inform you that the article has been submitted to the English Language Editing Services - MDPI, according to the certificate attached.

Thank you for your considerations,                                                     

Best regards,

Mariana Maroso Molina Irigaray.

Round 2

Reviewer 1 Report

Comments and Suggestions for Authors

No further comments

Comments on the Quality of English Language

To be improved

Reviewer 2 Report

Comments and Suggestions for Authors

The authors significantly improved their manuscript.